# Formation of Aggregate-Free Gold Nanoparticles in the Cyclodextrin-Tetrachloroaurate System Follows Finke–Watzky Kinetics

**DOI:** 10.3390/nano12040583

**Published:** 2022-02-09

**Authors:** Yuri Sergeyevich Pestovsky, Teerapol Srichana

**Affiliations:** 1Drug Delivery System Excellence Center, Prince of Songkla University, Songkhla 90110, Thailand; 2Genetic Engineering Department, Center for Research and Advanced Studies of the National Polytechnic Institute (Cinvestav), Irapuato 36824, Mexico; 3Faculty of Pharmaceutical Sciences, Prince of Songkla University, Songkhla 90110, Thailand

**Keywords:** gold nanoparticles, cyclodextrin, tetrachloroaurate reduction, green chemistry, Finke–Watzky kinetics, drug delivery vehicle

## Abstract

Cyclodextrin-capped gold nanoparticles are promising drug-delivery vehicles, but the technique of their preparation without trace amounts of aggregates is still lacking, and the size-manipulation possibility is very limited. In the present study, gold nanoparticles were synthesized by means of 0.1% (*w*/*w*) tetrachloroauric acid reduction with cyclodextrins at room temperature, at cyclodextrin concentrations of 0.001 M, 0.002 M and 0.004 M, and pH values of 11, 11.5 and 12. The synthesized nanoparticles were characterized by dynamic light scattering in both back-scattering and forward-scattering modes, spectrophotometry, X-ray photoelectron spectroscopy, transmission electron microscopy and Fourier-transform infrared spectroscopy. These techniques revealed 14.9% Au^1+^ on their surfaces. The Finke–Watzky kinetics of the reaction was demonstrated, but the actual growth mechanism turned out to be multistage. The synthesis kinetics and the resulting particle-size distribution were pH-dependent. The reaction and centrifugation conditions for the recovery of aggregate-free nanoparticles with different size distributions were determined. The absorbances of the best preparations were 7.6 for α-cyclodextrin, 8.9 for β-cyclodextrin and 7.5 for γ-cyclodextrin. Particle-size distribution by intensity was indicative of the complete absence of aggregates. The resulting preparations were ready to use without the need for concentration, filtration, or further purification. The synthesis meets the requirements of green chemistry.

## 1. Introduction

Cyclodextrins are cyclic oligosaccharides that consist of glucopyranose units and are widely applied in drug delivery due to their ability of drug encapsulation. This complexation protects drugs from decomposition, increases their solubility, and improves bioavailability. Inclusion complexes can release drugs in a sustained manner or enhance the release of poorly water-soluble drugs from the vehicles [1]. Several methods for the preparation of cyclodextrin nanoparticles with encapsulated drugs have been reported [2]. However, they require either the chemical modification of cyclodextrin or the use of organic solvents. Moreover, in the latter case the drug loading occurs during nanoparticle formation, and the exact techniques should depend on the drug and thus have limited versatility.

Gold nanoparticles are excellent carriers for drug delivery due to their tunable size, narrow size distribution, low toxicity, and easy synthesis and modification. They can be used with various payloads ranging from antibiotics and anticancer drugs to proteins and nucleic acids [3]. Unlike multiwall carbon nanotubes [4], gold nanoparticles can penetrate plant cells [5]. This possibility paves the way for their use as pesticide carriers as well [6], although it should be noted that the tested plants were different [4,5]. Conventional methods for the synthesis of gold nanoparticles are now being substituted by green methods that are based on tetrachloroaurate reduction by biomolecules. Green methods are not only eco-friendly but also lead to nanoparticle preparations that are free of any chemical admixtures. Starch is used in one of these methods [3].

We have recently synthesized gold nanoparticles using cyclodextrins [7,8]. Since cyclodextrins are produced by means of the enzymatic conversion of starch [1], this method is also a green one. Moreover, the ability of cyclodextrin-capped gold nanoparticles to encapsulate and release various drugs can be used beyond drug delivery. Gold nanoparticles with immobilized cyclodextrins could be used for the sequestration of antibiotics with their subsequent release into the nanocomposite-containing medium for degradation [9,10]. Being reusable and easily separable by centrifugation, these nanoparticles would be useful for the preconcentration step, especially when pH adjustment is needed. For example, in the case of the sonochemical degradation of cefazolin sodium, the performance of the nanocomposite in the alkaline solution was suboptimal [9]. Furthermore, this technique is not limited to antibiotics; it can even be extended to the photocatalytic degradation of phenol [11].

The interaction of 0.003 M native β-cyclodextrin with 2.86 × 10^−4^ M HAuCl_4_ in phosphate-buffered saline was shown to be possible at pH 6–8, but monodispersed nanoparticles were produced only at pH 7–8. The absorbance at maximum was about 0.8. The synthesis was carried out at 100 °C because monodispersity was lost at lower temperatures; therefore, temperatures lower than 70 °C were not investigated. Two other cyclodextrins yielded slightly elongated nanoparticles that had a small trailing over 750 nm in the absorbance spectra [12]. A similar technique involving 0.002 M β-cyclodextrin and 2 × 10^−4^ M HAuCl_4_ in a buffer at pH 7 upon heating at 100 °C under stirring yielded uniform nanoparticles of 14–21 nm. Interestingly, in the beginning of the reaction the nanoparticles were much larger than at the end [13]. All three native cyclodextrins at 0.0016 M, upon interaction with 2 × 10^−4^ M HAuCl_4_ in alkaline solution under microwave irradiation, produced monodispersed gold nanoparticles with a maximum absorbance in the range of 0.8–1.15 [14]. An optimal pH of 10.5 was reported for the formation of gold nanoparticles under microwave irradiation in a reaction mixture containing 0.002 M β-cyclodextrin and 10^−4^ M HAuCl_4_ at pH values of 9.5–11.5. The maximum absorbance was about 0.8 [15]. No attempt to purify them was made [14,15]. In contrast to this, the β-cyclodextrin-epichlorohydrin polymer can reduce tetrachloroauric acid at a pH of about 3 [16].

However, in all of the above-cited publications, the nanoparticles were analyzed by dynamic light scattering in the back-scattering mode only. Therefore, the complete absence of aggregates was not ensured. Indeed, the above investigations [12,13,14,15] were aimed at biosensing applications and not at drug delivery. However, an aggregation of nanoparticles in the blood vessels is known to cause embolism [17,18]. Hence, the absence of aggregates is critical for drug-delivery applications.

In contrast, some of our previous samples that were prepared using all three cyclodextrins were aggregate free. Moreover, we demonstrated the possibility of carrying out the synthesis at room temperature without the need for stirring [7,8]. However, the size of the gold nanoparticles synthesized by means of green methods usually cannot be manipulated [19,20,21,22,23,24]. We attempted to introduce some size control by varying the reaction duration [8]. This possibility was actually rather limited. In the present study, we investigate another powerful factor—pH.

Despite numerous reports of the instability of α- [25,26] and β-cyclodextrins [25,26,27] in acid media, their hydrolysis in alkaline media has never been reported. Moreover, the heating of α- or β-cyclodextrin in KOH at 100 °C for 90 min did not decrease the concentrations [25]. Therefore, we attempted to make the solution more alkaline in order to increase cyclodextrin ionization, thus facilitating its coordination with gold cations.

In the case of the citrate method, the particle sizes and especially their distribution are pH-dependent but only a slight temperature dependence of these parameters was observed at 70 °C, 85 °C, and at boiling point. In contrast to low pH values, high pH values yielded small and large nanoparticles that coexisted in the reaction mixture, which implied a different nucleation or coagulation step. The optimal pH at 70 °C was 7.5. The introduction of pH control and a reduced molar ratio of citrate to tetrachloroaurate made it possible to produce gold nanoparticles at a much higher concentration compared to the classical Frens method while maintaining stability and low polydispersity [28].

An alkaline solution is favorable not only for cyclodextrin ionization but also for tetrachloroaurate hydrolysis. The redox potentials of tetrachloroaurate and tetrahydroxoaurate are 1.00 V and 0.7 V, respectively, while those of partially hydrolyzed tetrachloroaurate ions are unknown but should be somewhere in between [29]. The increase in the average size of gold nanoparticles synthesized in the presence of sodium benzenesulfonate at high pH has been ascribed solely to the effect of tetrachloroaurate hydrolysis [30]. A third factor that should be taken into account is the ionic strength that increases with the alkalinity of an unbuffered solution and promotes the aggregation of nanoparticles and nanoclusters. Therefore, particle-size distribution in the cyclodextrin–tetrachloroaurate system is the result of a complex interplay between three factors: cyclodextrin ionization, tetrachloroaurate hydrolysis, and ionic strength. In turn, all of these parameters are pH dependent.

Hence, the goal of the present study was to characterize gold nanoparticles that were produced by means of tetrachloroauric acid reduction with cyclodextrins at several pH values at room temperature. It allowed us, for the first time, to propose a ready-to-use technique for the one-step green synthesis of cyclodextrin-capped gold nanoparticles with different size distributions and without aggregates or other admixtures in the resulting preparations. The reaction was shown to follow Finke–Watzky kinetics. These aggregate-free gold nanoparticles with immobilized cyclodextrins are promising drug-delivery vehicles.

## 2. Materials and Methods

### 2.1. Materials

Sodium hydroxide and HAuCl_4_ were from Sigma-Aldrich (Saint Louis, MO, USA). Cyclodextrins were from Wacker Chemie AG (Bracknell, UK). Sodium chloride was purchased from VWR International Ltd. (Poole, UK). Water (18.2 MΩ/cm) was of type 1, purified by ELGA Purelab Chorus 1 (ELGA, High Wycombe, UK).

### 2.2. Synthesis of Gold Nanoparticles under Reflux Conditions

A solution volume of 27,187 µL of 0.004 M β-cyclodextrin with a calculated amount of NaOH in a 50 mL glass, custom-made, pear-shaped, double-necked flask was mixed for 10 min using a Vortex-Genie 2 G-560 (Scientific Industries, Inc., Bohemia, NY, USA). An aliquot of 2913 µL was then mixed with 87 µL of 0.1 M HAuCl_4_ prepared from the solid chemical and taken for pH adjustment to 10.56. The flask was put in a sand bath, covered with sand to the solution level, and reflux was started. When the solution began to boil, the reaction was started with 726 µL of 0.1 M HAuCl_4_. The concentration of HAuCl_4_ in the solution became 0.1% (*w*/*w*). The final volume of the solution was 25 mL. Reflux was continued for 26 min, and the solution was stored at 4 °C. Some black precipitate that was observed was discarded.

In the case of the Fourier-transform infrared (FTIR) spectroscopy sample preparation, the solution was centrifuged at 9000× *g* for 55 min at 4 °C in a fixed-angle rotor RA-508C (with adaptors) using centrifuge model 5922 (Kubota, Tokyo, Japan). The nanoparticles were resuspended in water, freeze dried and used immediately.

In the case of the X-ray photoelectronic spectroscopy (XPS) sample preparation, the solution was transferred to a 39 mL 25 × 89 mm quick-seal, round-top polyallomer tube (Beckman Instruments, Inc., Fullerton, CA, USA) together with a solution from another experiment in order to reach the volume of 39 mL and ultracentrifuged at 164,684× *g* for 26 min, including 6.5 min for acceleration (70 Ti fixed-angle rotor, Optima L-100 XP, Beckman Coulter, Fullerton, CA, USA). Preliminary experiments using an ordinary centrifuge with common 15 mL tubes at the maximal speed of the rotor (9170× *g*) demonstrated unacceptably low yields.

The bottom portion of the supernatant was transferred to a 31 mL open-top, thick-wall polypropylene tube 25 × 89 mm (Beckman Instruments, Inc., Fullerton, CA, USA) and centrifuged again at 41,171× *g* for 25 min (including 4 min for acceleration). The remaining unused solutions from the samples characterized by scanning electron microscopy (SEM, Appendix A) as well as from other kinetic experiments were mixed together and also centrifuged at 126,086× *g* in the same tube. Small amounts of supernatants could not be removed after centrifugation because of loose precipitates. The resulting precipitates were freeze dried, scratched out with a spatula, mixed together and stored in a desiccator.

### 2.3. Room-Temperature Synthesis of Gold Nanoparticles

A solution volume of 11,849 µL of 0.001 M, 0.002 M, or 0.004 M cyclodextrin with a calculated amount of NaOH (added in order to speed up the dissolution of cyclodextrin and facilitate subsequent titration) in a 15 mL plastic tube was mixed for 10 min at maximal speed. An aliquot of 2962 µL was then taken for pH adjustment. The reaction was started with 113 µL of 7.9% (*w*/*v*) HAuCl_4_ solution prepared from commercially available 30% (*w*/*w*) HAuCl_4_ in diluted HCl. The reaction mixture (9 mL, now containing 0.1% (*w*/*w*) HAuCl_4_) was immediately vortexed for 10 min at maximal speed. During the reaction, 1 mL aliquots were taken for analysis. They were immediately stored at 4 °C for overnight storage followed by centrifugation at 4 °C in a fixed-angle rotor AT-2018M. The speed and duration were separately selected for each sample. The precipitates were resuspended into 1 mL or 1.5 mL of water and the supernatants were discarded.

When the catalytic activity of chloride was being probed, the reaction mixture also contained 11.2 mg of NaCl. A preliminary experiment confirmed the stability of the nanoparticles at this concentration of NaCl.

In the case of 0.004 M γ-cyclodextrin, the solution volume was 4 mL. No centrifugation was used. After pH adjustment the solution was transferred to the cell and the process was monitored in real time with a UV-1800 spectrophotometer or a zetasizer.

### 2.4. pH Adjustment

37.6 µL of 7.9% (*w*/*v*) HAuCl_4_ were added to 2962 µL of the solution followed by vortexing for 10 min at maximal rate. A SevenCompact pH meter (Mettler-Toledo AG, Shanghai, China) equipped with an InLab Viscous Pro-ISM electrode (Mettler-Toledo GmbH, Greifensee, Switzerland) was used. The pH was then adjusted using NaOH solution while mixing. It should be noted that the pH of the solution may slowly decrease upon mixing, but only stable pH values were treated as final values. The electrode was stored in 3 M KCl (Mettler Toledo InLab Solutions, Mettler Toledo GmbH). Immediately before use it was washed with water and wiped. The consumed amount of NaOH that was needed to adjust the pH of the main solution was then calculated. Therefore, the solution used for the synthesis of the nanoparticles was not contaminated with KCl coming from the electrode.

### 2.5. Freeze Drying of Nanoparticles

The samples were frozen at −20 °C and put into a freeze dryer (Lyovapor L-300 Pro, BÜCHI, Flawil, Switzerland) with the following ice-condenser parameters: pressure 0.021 mbar and temperature −105 °C. Primary drying was carried out for 21 h at −40 °C followed by 7 h of secondary drying from 10 °C to 25 °C. The freeze-dried samples of nanoparticles intended for stability test were stored at −20 °C. They were briefly incubated at 50 °C followed by the addition of 1 mL of water and mixing for 10 min at about 1280 rpm.

### 2.6. Characterization of Nanoparticles

A one-beam GENESYS 6 spectrophotometer (Thermo Spectronic, Rochester, NY, USA) and double-beam UV-1800 spectrophotometer (Shimadzu, Kyoto, Japan) were employed, and the samples were in 1 mL quartz cells. Water was used as a blank. The former spectrophotometer was connected to a computer running VISIONlite™ software (version 1.0). The absorbance spectra acquired by this spectrophotometer were smoothed after having been imported to UVProbe software (version 2.43). Spectra acquired by the UV-1800 were smoothed at the first order using its embedded software. The scaling factor was 1, and the delta lambda was 5.000. The spectra were plotted by UVProbe 2.43. Samples with a maximal absorbance exceeding 1.1 were diluted and the resulting spectra were multiplied by the dilution factor.

The zeta potential and particle-size distribution were measured using a Zetasizer Nano ZS (Malvern Instruments Ltd., Malvern, UK) equipped with 633 nm ‘red’ laser and a DTS1060 or DTS1070 folded capillary cell from the same manufacturer. The measurements were made at 25 °C in a dual-angle mode. Three measurements were performed for each sample. If multiple scattering occurred, the samples were diluted as necessary. The refractive index was chosen according to [8] for the intensity size-distribution peak. Peaks corresponding to a particle size of 100 nm or above were ignored, provided that other ones existed. For peaks absent from [8], the ‘default’ values of refractive index of gold nanoparticles (n = 0.2, k = 3.32) were used. The data from the measurements corresponding to the same sample were averaged. The results from the measurements reflecting only peaks corresponding to particle sizes of more than 100 nm were excluded from averaging, provided at least one measurement had revealed a peak below 100 nm for the same sample. Aggregate-free samples were defined as those displaying no objects larger than 100 nm in the size distribution by the intensity in both back-scattering and forward-scattering modes in all of the six runs per sample. In the case of zeta-potential distributions, the main peak in the average-number size distribution was used for data correction. Zeta-potential distribution data from the three measurements corresponding to the same sample were also averaged.

The calculations were performed using Zetasizer Software (version 7.13). The general-purpose model was used for data processing. Zeta-potential data were processed in auto mode using the Smoluchowski model.

The yield was calculated as follows. The concentration of nanoparticles was calculated using Beer’s law and the extinction coefficient of gold nanoparticles at 550 nm corresponding to their main-number size-distribution peak as calculated by means of a calibration curve [31]. Number-size distribution was assumed to be monomodal regardless of whether it was actually the case. The mass of a nanoparticle having a given diameter was calculated assuming that the nanoparticle was a perfect sphere and that its density was the same as that of bulk gold (19.3 g/cm³). From the calculated mass, the amount of gold in the nanoparticle was calculated. Multiplication by the number of nanoparticles in the volume of solution gave the total amount of gold atoms. Division of this value by the amount of tetrachloroauric acid at zero time gave the yield. For γ-cyclodextrin, the particle diameter was calculated from the absorbance spectrum by means of a calibration curve [31].

Transmission-electron-microscope (TEM) images were acquired by means of a JEM-2010 (JEOL, Tokyo, Japan) equipped with an Olympus CCD camera. The voltage was 200 kV, and the spot size was 3 mm. The solutions were dripped onto a 200-mesh carbon-coated copper grid (EMS, Hatfield, PA, USA) and placed in a desiccator. The solutions at the early stages of the reaction were not centrifuged prior to TEM sample preparation.

For XPS, a beamline 3.2Ua at the Synchrotron Light Research Institute, Thailand [32] was used. A mixed and freeze-dried sample as described in Section 2.2 was pressed into a 5 mm pellet and attached to the holder with conductive carbon tape. It was then placed in the load-lock chamber for vacuum pumping. The analysis was carried out in the analysis chamber. The incoming photon energy from the beamline was set to 650 eV and focused at a 1 mm spot with an incident angle of 70° at room temperature. The energy of the emitted electrons was measured by means of a concentric hemi-spherical analyzer (CLAM2, Thermo VG Scientific, London, UK) with a pass energy of 50 eV. A survey-scan spectrum with an energy step of 1 eV was measured in order to inspect the elements on the sample surface. Narrow scans of the elements visible in the survey scan were measured using an energy step of 0.1 eV. For the binding-energy-scale calibration, the C1s peak (285 eV) originating from the adventitious carbon was used. The measurement was conducted in an ultrahigh-vacuum chamber at 7.6 × 10^−9^ mbar.

The core-level spectra in the narrow scans were analyzed by the Visual Basic Application code in Microsoft Excel 365 for curve fitting. The peak area fitted by the Gaussian peak and normalized by the atomic-sensitivity factor was used to evaluate the atomic concentration of each element on the sample surface.

FTIR spectra were acquired for 64 scans by means of a PerkinElmer Spectrum One FT-IR Spectrometer (Buckinghamshire, UK) connected to a computer running Spectrum v5.0.2 software. A freeze-dried sample of β-cyclodextrin or gold nanoparticles (centrifuged and resuspended into water, Section 2.2) was ground with KBr (heated overnight at 90 °C) in a mortar and pressurized into a pellet. The spectra were normalized and plotted by means of Spekwin32 1.72.0.

For the nuclear-Overhauser-effect-spectroscopy (NOESY) experiment, a volume of 9 mL of a solution containing 0.1% (*w*/*w*) HAuCl_4_ and 0.004 M α-cyclodextrin was prepared. A volume of 3 mL of this solution was consumed for pH adjustment to 10.56. To the remaining 6 mL, a calculated amount of NaOH solution was poured, and after mixing for 10 min at maximal speed the solution was freeze dried. The precipitate was reconstituted in 700 µL of heavy water immediately before NMR-spectra acquisition.

The NOESY spectra were acquired using a 500 MHz NMR spectrometer Unity Inova (Varian, Palo Alto, CA, USA) with the following parameters: relaxation delay 2 s, 128 evolution time increments, number of transients 16, Fourier number 1024, 1024 data points. Only Gaussian weighing was used.

When rotating-frame nuclear-Overhauser-effect spectroscopy (ROESY) or one-dimensional NMR spectroscopy was employed, initially, the preliminary solution containing 43.3 mg of α-cyclodextrin, 30.8 µL of 30% (*w*/*w*) HAuCl_4_ (in diluted HCl), and 2969 µL of heavy water was prepared and its pH was adjusted to 6.15. An attempt to titrate the solution to a higher pH resulted in the immediate appearance of a black precipitate, which probably consisted of aggregated gold particles. A volume of 775 µL of heavy water (or deuterated dimethyl sulfoxide [DMSO]) and a premix of 17.06 µL of NaOH in D_2_O and 8.24 µL of 30% HAuCl_4_ in diluted HCl were added to 11.58 mg of α-cyclodextrin immediately before NMR-spectra acquisition. In the control experiments, α-cyclodextrin was dissolved in 800 µL of heavy water or DMSO without the addition of the above premix. The pH of the control solution was 6.16 without adjustment.

The ROESY experiment was run on CryoProbe Prodigy 5 mm BBO 500 MHz, AVANCE NEO, ASCEND 500 MHz NMR Spectrometer (Bruker, Fällanden, Switzerland). Data were processed on TopSpin NMR software (version 4.11). The NMR experiment was carried out at 298 K, pulse program: roesyphpp.2, 16 scans. The ROESY spectra were acquired with standard parameters set by Bruker for compound geometry (pulse program roesyphsw). Each spectrum consisted of a matrix of 2K (F2) by 256 (F1). Relaxation delay was 2 s, and spin-lock mixing time (p15 pulse) was set to 200 ms. The size of free-induction decay covered 4000 Hz. Gaussian apodization functions were applied in both dimensions.

## 3. Results and Discussion

We have previously synthesized gold nanoparticles using 0.001 M γ-cyclodextrin at pH 10.77 and have had failed attempts at 0.002 M and 0.004 M γ-cyclodextrin at pH 10.56 [8]. In order to address this discrepancy, we carried out the experiment using 0.001 M γ-cyclodextrin at pH 10.56 (Appendix A). However, we recovered only aggregates and precipitate that were visible to the naked eye, despite an apparently high zeta potential (Appendix A). The pH of the solution dropped to 7.33 after the reaction. Therefore, our previous success at this concentration could be completely explained by the higher pH value, thus underlining the importance of pH control and the need for investigation into the reaction system at higher pH values.

At pH 12, most samples that were prepared using α-cyclodextrin aggregated and precipitated out (Appendix A). After almost 12 h, soluble aggregates accumulated at a rather high concentration. They tended to precipitate during storage. After the reaction, a thin film of metal gold that appeared as a ‘golden mirror’ formed on the walls of the tube. Therefore, nucleation had begun to proceed on the walls of the tube instead of the seeds in solution. The absence of seeds, in turn, could be explained by the almost complete conversion of [AuCl_4_]^−^ to [Au(OH)_4_]^−^ at high pH. The latter species cannot initiate nucleation. Small amounts of formed nanoclusters aggregated because of high ionic strength and then precipitated out, thus becoming unable to act as seeds. Dynamic light scattering still revealed free nanoparticles in most samples (Appendix A). In the first two samples, nanoparticles smaller than 2 nm coexisted with larger particles (Appendix A). The reaction with β-cyclodextrin ended up in almost complete aggregation as revealed by the absorbance spectra (Appendix A) and by the naked eye. However, free nanoparticles could be revealed by dynamic light scattering at any time point (Appendix A). If the sample was withdrawn at an intermediate time, soluble aggregates presented a blue color to the solution and could be recovered, but they precipitated during storage. Zeta-potential data confirmed the stability of the system (Appendix A), but they actually corresponded to soluble aggregates and a small fraction of remaining free particles. When γ-cyclodextrin was used, real-time monitoring with a spectrophotometer (Appendix A) did not reveal the absorbance maximum of gold nanoparticles even after a prolonged time. Instead, the solution gradually became yellow. This color might correspond to the formation of gold nanoclusters [33]. However, the zetasizer revealed mostly aggregates in the forward-scattering mode (Appendix A). In the back-scattering mode, free nanoparticles were continuously revealed together with aggregates, but their size did not increase with time (Appendix A). The zeta potential of nanoparticles (−29.7 ± 1.1 mV) decreased after the dilution of the sample (Appendix A), indicating the possible breakdown of the aggregates.

Obviously, the nanoparticles prepared at pH 12 cannot be used for drug delivery due to the presence of aggregates and even of an insoluble precipitate. Aggregation was caused by the high ionic strength of the reaction mixture due to the high concentration of NaOH needed to reach pH 12. On the other hand, our previous results implied that the pH should not be below 10.56 [7,8]. We discovered a batch of brown ‘β-cyclodextrin’ that could immediately reduce tetrachloroauric acid at room temperature, even at pH 4.97, but this chemical was actually not β-cyclodextrin, as confirmed by its different absorbance spectrum. Since the optimal pH should be somewhere between 10.56 and 12, we tried pH 11.5.

For the 0.001 M α-cyclodextrin, an induction period of 51 h was observed (Appendix A). Most samples aggregated (Appendix A) and the precipitate was seen by the naked eye. Although nanoparticles could still be recovered at a low concentration, they were of no value because of the precipitate. However, the zeta-potential values of the recovered nanoparticles indicated they could be predicted to be stable (Appendix A). The explanation of this discrepancy is that the unstable particles had precipitated out well before the measurements. The low concentration of cyclodextrin was possibly the cause of the instability and the inability to form a monolayer on the particle surface.

In contrast to our previous results at pH 10.56 [8], at pH 11.5 aggregate-free gold nanoparticles at a high concentration could be produced using 0.002 M α-cyclodextrin (Appendix A). The induction period was 7 h. After the rapid-accumulation stage, further size change over time was negligible and the samples were monodispersed. The only exception was the final sample that contained nanoparticles of 2 nm (Appendix A) and 45 nm (Appendix A) as well as aggregates. We have selected this successful result to check our previous hypothesis of the stabilization of the gold intermediate [AuCl_2_]^−^ by chloride because this assumption may lead to the conclusion that chloride can act as a catalyst. Commercially available HAuCl_4_ that was used in all of the experiments, including the one described above, was supplied as a solution in the form of diluted HCl. Indeed, the reaction proceeded much slower when a preparation of HAuCl_4_ without HCl was used (Appendix A). During the titration of the reaction mixture before the start of the reaction, the overconsumption of NaOH was noticed, which pointed out the extensive hydrolysis of tetrachloroaurate. Another interesting observation was that the sample isolated from the reaction mixture 8 h after the start but kept at room temperature was rather different from the freshly isolated one. At a very low concentration of the active species [AuCl(OH)(H_2_O)_2_]^+^, the probability of nucleation seemed to depend on the volume of the solution, thus causing irreproducibility. As observed from the absorbance spectra (Appendix A), the final sample reached a very high concentration of nanoparticles. However, by that time an island-type film had appeared on the walls of the tube. Therefore, the nanoparticles in solution formed from secondary seeds that detached from this film, and only one-fourth of the seeds appeared in a homogeneous process in solution. Surprisingly, the absence of chloride also promoted aggregation. In contrast to the aggregate-free samples produced in the presence of chloride, all of these samples contained aggregates.

In most experiments, the reaction mixtures in contact with the electrode of the pH meter developed a red color more rapidly than the corresponding main solutions, probably because of trace amounts of KCl coming from the electrode. Nevertheless, in the specially designed experiment carried out in the presence of additional chloride, which was introduced as NaCl, the reaction did not accelerate (Appendix A), perhaps because the employed concentration of NaCl did not match the unknown concentration of KCl in the above-mentioned mixtures. The induction period became 12 h. During this time the solution gradually darkened. It seemed that the chloride accelerated nucleation (Appendix A) as evidenced by a solution-darkening time of less than 1 h compared to 4.5 h in the absence of the added NaCl. The formed nanoclusters rapidly evolved to large nanoparticles (Appendix A and Figure 1a). The nanoparticles appeared to be highly unstable upon centrifugation and were, therefore, mainly lost. Finally, more-stable nanoparticles appeared by detaching from the formed island-type film, as in the experiment without chloride, and reached a reasonable concentration. The film later turned into a golden mirror. Zeta-potential distributions in the absence of chloride (Appendix A), at the usual concentration of chloride (Appendix A), and with excess chloride (Appendix A) were notably different as well.

For 0.004 M α-cyclodextrin, the induction period was much longer and the concentration of nanoparticles was somewhat higher compared to pH 10.56 [8] (Appendix A), although some nanoparticles were still lost with the supernatant during centrifugation. The particle-size distribution was bimodal (Appendix A) and different from pH 10.56 [8]. Nanoclusters were often found to coexist with nanoparticles (Appendix A). Most of the samples were aggregate-free, despite the wide zeta-potential distribution (Appendix A). Unfortunately, sample instability during freeze drying was observed. The resulting black precipitate failed to be resuspended in water. Although free nanoparticles with altered size distribution and aggregates were still revealed by dynamic light scattering, their concentration was below the detection limit of the spectrophotometer.

For 0.004 M β-cyclodextrin, the reaction proceeded rapidly but a little slower than at pH 10.56 and a 2.5-times-higher concentration of nanoparticles could be reached (Appendix A). From 2 h onwards, the samples were violet but looked red in transient light. The size distribution became bimodal (Appendix A) and different from that registered at pH 10.56 [8]. Some samples were aggregate free. This observation could be verified by zeta-potential data (Appendix A). TEM images revealed monodispersed, polycrystalline, spherical nanoparticles, some of which were fused (Figure 1c).

For 0.001 M γ-cyclodextrin, the reaction at pH 11.5 was much slower than at pH 10.77 [8]. The resulting concentration of nanoparticles was significantly lower and broadened absorbance peaks were noticed (Appendix A). Indeed, dynamic-light-scattering data (Appendix A) revealed that all samples were contaminated with aggregates, despite the good zeta-potential distribution (Appendix A).

Unlike the failed outcome at pH 10.56 [8], at pH 11.5 gold nanoparticles could be produced using 0.002 M γ-cyclodextrin at a high concentration after the induction period of 8 h (Appendix A). Comparisons between the back-scattering (Appendix A) and forward-scattering data (Appendix A) revealed a bimodal distribution. Unfortunately, all of the samples contained aggregates that were hard to be revealed by zeta-potential-distribution data (Appendix A). For 0.004 M γ-cyclodextrin, in contrast to the data acquired at pH 10.56 [8] and pH 12.12, at pH 11.5 the absorbance peak of gold nanoparticles appeared shortly after the start and increased with time. The absorbance exceeded 6 after overnight incubation (Appendix A). As in the case of 0.002 M γ-cyclodextrin, a bimodal size distribution (Appendix A) and aggregates were observed. The zeta-potential distribution was very wide and had an unusual shape (Appendix A). It should be underscored that the samples had not been centrifuged; however, aggregates still appeared, probably because of the high ionic strength due to the high concentration of NaOH. Nanoparticles present even after 17 min from the start and the absence of nanoclusters and of clear kinetics demonstrated the complexity of the particle-formation mechanism.

For 0.001 M α-cyclodextrin at pH 11, extensive aggregation was again observed. Number size distributions of early samples in back-scattering mode revealed peaks of particles less than 1 nm in size, which were gold nanoclusters that later grew to 3 nm and finally to 5 nm (Appendix A). An absorbance spectrum of the final sample demonstrated a high concentration of nanoparticles with a broad size distribution of mostly aggregates (Appendix A). Indeed, dynamic light scattering revealed nanoparticles of various sizes from 2 nm to 190 nm (Appendix A). The pH of the solution dropped to 8.27 after the reaction. Interestingly, the zeta-potential values again predicted the stability of the system (Appendix A), but these objects were actually soluble aggregates and not free nanoparticles.

For 0.002 M α-cyclodextrin, the reaction was even slower than at the pH of 11.5, with an induction period of 15 h. The final concentration of nanoparticles was higher (Appendix A), and most of the samples had a bimodal size distribution (Appendix A). In three samples, nanoclusters were revealed (Appendix A). Most of the nanoparticles were nearly spherical, but other shapes and fused particles were also present (Figure 1e). The nanoparticles were polycrystalline and only one single crystal was found in TEM images, but all of the samples contained aggregates despite the narrow zeta-potential distribution of some of them (Appendix A).

For 0.004 M α-cyclodextrin, the reaction proceeded a little more rapidly than at pH 11.5, and the concentration of nanoparticles significantly exceeded those registered at this concentration of α-cyclodextrin at other pH values (Appendix A). In order to recover the nanoparticles, the centrifuge should be run at maximal speed (20,630× *g*). The particle size (Appendix A) was almost the same as at pH 10.56 [8]. They were monodispersed and mainly polycrystalline. Fused nanoparticles were common. Single crystals were also present, sometimes even fused together and with polycrystalline nanoparticles (Figure 1g). Unfortunately, all of the samples, even when centrifuged at 1470× *g* with high losses, contained aggregates, although this conclusion is not obvious from the zeta-potential-distribution data (Appendix A).

For 0.004 M β-cyclodextrin at pH 11, the reaction kinetics were the same as at pH 11.5. Extensive efforts to avoid aggregation during centrifugation decreased the yield but the concentration of nanoparticles (Appendix A) was still much higher than at pH 10.56 [8]. However, not all of the samples were aggregate free, and unusual zeta-potential distributions were common (Appendix A). Some of the samples still had a bimodal size distribution, while the back-scattering (Appendix A) and forward-scattering (Appendix A) measurements of other samples yielded the same results. Most of the nanoparticles were polycrystalline, although single crystals were also found. Fused nanoparticles were rather widespread. Moreover, two fused nanoparticles could look like a single crystal (Figure 1h).

For 0.002 M γ-cyclodextrin, the nanoparticles could be produced after an induction period of 23 h (Appendix A). The pH dropped to 10.61 after the reaction. In striking contrast to the samples prepared with other cyclodextrins, the TEM images revealed that most of the formed nanoparticles had an irregular shape (Figure 1j). The last sample had a very high concentration of nanoparticles with a bimodal size distribution (Appendix A). Unfortunately, this sample failed to be reconstituted after freeze drying. It displayed a black precipitate and a solution with zero absorbance, although free nanoparticles and aggregates were revealed by dynamic light scattering. The zeta potential remained the same (Appendix A). For 0.004 M γ-cyclodextrin, the reaction proceeded much more rapidly than at pH 11.5, which was easy for real-time monitoring (Appendix A). After 2 h, the spectrophotometer became overloaded due to the very high concentration of nanoparticles. Therefore, further monitoring was carried out in diluted solutions and the resulting spectra were multiplied by the dilution factors of 5 or 10. The reaction proceeded under these conditions without a broadening of the spectra, which indicated no aggregation could be seen.

From the absorbance spectra, the reaction had three distinct stages: (1) the induction period, (2) the rapid stage (within minutes), and (3) plateau. Therefore, it followed Finke–Watzky kinetics [34,35], assuming the slow growth of gold nanoclusters to a critical size and subsequent autocatalytic growth to gold nanoparticles. This mechanism provides a simple explanation of the growth of nanoparticles that almost stopped at the plateau stage due to the depletion of reagents. Indeed, the XPS spectra of the gold nanoparticles did not reveal Au^3+^ (Figure 2b). Moreover, single-crystal nanoparticles found in the TEM images should have evolved from single nanoclusters, thus providing further proof of the applicability of the Finke–Watzky mechanism to tetrachloroaurate reduction with cyclodextrins.

However, Finke and Watzky state that their mechanism is general, which is applicable to various processes. Therefore, it often turns out to be a simplification in particular cases [35], of which our system was no exception. Most nanoparticles found on TEM images were not single crystals, and the TEM images of our early samples did not reveal gold nanoclusters (Figure 1a,b,d,f). Instead, large nanoparticles of irregular or sometimes spherical shapes were detected. Most of them were polycrystalline, although single-crystal nanoparticles were also found. Fused nanoparticles were present as well. Magnified images revealed that the large objects were not aggregates. They were especially frequent when β- or γ-cyclodextrin was used. The appearance of large nanoparticles in the beginning was independently confirmed by another research group [13]. In the sample prepared with 0.002 M α-cyclodextrin, rounded triangles and squares that were absent in other samples were found. Hence, the mechanism is more complex. We have previously stated [8] that the formation of gold atoms is preceded by tetrachloroaurate hydrolysis and protonation, cyclodextrin deprotonation, and reduction of [AuCl(OH)(H_2_O)_2_]^+^ to [AuCl_2_]^−^.

The absence of the alkaline hydrolysis of cyclodextrins [25] means that the mechanism of Au(III) reduction does not involve cyclodextrin ring opening. Instead, it involves the oxidation of the deprotonated hydroxyl groups of cyclodextrin. Variable-pH NMR-spectroscopy data have not shown β-cyclodextrin deprotonation at a pH below 12. Its secondary hydroxyl groups deprotonate more readily than the primary ones and have pK_a_ values of 13.5 ± 0.2 [36]. In the complex of Na_3_[Na_3_Cu_3_(α-cyclodextrin)_2_]·acetone·32H_2_O, both copper and sodium were coordinated to secondary hydroxyls, one of them (O2) being deprotonated [37]. However, the β-cyclodextrin-epichlorohydrin complex reduced tetrachloroaurate by its primary hydroxyl [16]. The preferential oxidation of the primary hydroxyl group was also concluded for the case of tetrachloroaurate reduction by native β-cyclodextrin from density-functional-theory calculations, which showed the binding of the resulting carboxyl group to the particle surface. Interestingly, gold atoms in the proximity of this group were found to have a charge from +0.2 to +0.3 e [12].

It should be noted there might be an alternative method of cyclodextrin activation. The vapor diffusion of methanol into a solution containing γ-cyclodextrin and NaOH was reported to result in the formation of a metal-organic framework having the composition of [(NaOH)_2_·(γ-cyclodextrin)]_n_. In this framework, sodium cations coordinate with the secondary hydroxyl groups of γ-cyclodextrin and with unidentified moieties, which may be either water molecules or OH^−^ ions. Framework formation was not accompanied by cyclodextrin deprotonation [38]. However, it is unknown whether the coordination of NaOH to γ-cyclodextrin influences its redox potential and even whether this coordination occurs in solution. Moreover, there are no data on other cyclodextrins.

The reduction of [AuCl(OH)(H_2_O)_2_]^+^ to [AuCl_2_]^−^ is dielectronic, and it is unclear whether it involves the formation of an unstable Au(II) intermediate. According to density-functional-theory calculations, the Au(II)-disproportionation reaction is thermodynamically favorable in an aqueous solution for both Cl^−^ and H_2_O as ligands. There are no data on OH^−^ but it was found to be exergonic for all of the other investigated ligands except for CO. Its driving force is the high solvation energy of the trivalent gold ion. For the chloride ligand associative mechanism, in other words, the formation of a transition state [Au_2_Cl_6_]^2−^ was found to have the smallest activation barrier; its gold atoms are connected through one chloride ligand [39].

However, all of the above details do not contradict the Finke–Watzky mechanism. Adding any number of fast-reversible stages prior to its two steps makes them pseudo-elementary, without altering the kinetics [35] or the main conclusions. Therefore, there should be another reason for the preferential formation of polycrystalline nanoparticles. We have hypothesized that they originate from the aggregation of the nanoclusters [8]. The seed with a critical size, thus being able to grow in an autocatalytic way, seems to appear more easily through the aggregation step rather than through the growth of the initially formed nanocluster, because the latter growth has positive enthalpy due to surface tension [34].

The average size of the primary nanocluster before aggregation might be determined not only by thermodynamics but also by the distance and relative position of the freshly formed gold atoms. [Au(CN)_4_]^−^ anions can self-aggregate by intermolecular interactions between gold and cyanide when the Lewis acidity of gold increases because of the presence of hydrogen bonds or metal cations. Aurophilic interactions were not observed for these anions [40]. The aggregation of [AuCl_4_]^−^ and other gold ions with chloride ligands has not been observed. Instead, the pattern of coordination of the gold ions with the single cyclodextrin molecule might play a role.

Being inspired by the previous reports on the observation of [Au(CN)_4_]^−^ [40] and coordinated water [41,42] by NMR, we have tried this powerful method on our system. Gold nanoclusters encapsulated in α-cyclodextrin have been successfully investigated by ROESY, but, unlike ours, they were capped with glutathione. One of its protons was easily sensed, which provided the structural insight [43].

To the best of our knowledge, the gold ions present in our system, such as [AuCl(OH)_3_]^−^ and [AuCl(OH)(H_2_O)_2_]^+^, have never been studied by NMR. The gold atom itself is not suitable for NMR [44]. Therefore, we have acquired ¹H NMR spectra. The assignment of the signals of α-cyclodextrin itself was based on the literature data [45,46]. Data for gold are available only for complexes with organic ligands. The available information for Au(I) is as follows. The complex (^Ad^CAAC)AuOH, where ^Ad^CAAC is N-substituted 2-adamantylalkylaminocarbene, has a broad singlet at −0.29 ppm (in C_6_D_5_Br) corresponding to the hydroxyl [47]. The hydroxyl group in the substance (IPr)AuOH, where IPr corresponds to 1,3-bis(2,6-diisopropylphenyl)imidazol-2-ylidene, had a resonance at −0.26 ppm in C_6_D_6_ [48]. As for Au(III), the coordinated water in a complex cation had a singlet at 5.33 ppm (in CD_2_Cl_2_), other ligands being 2-phenylpyridine and 1,3,5-tris(trifluoromethyl)benzene [49]. Coordinated hydroxyl in [Au_2_(OH)_2_(tppz)]Cl_4_, where tppz is 2,3,5,6-tetrakis(2-pyridinyl)pyrazine, corresponds to a broad singlet at 6.01 ppm in deuterated DMSO. Its chemical shift was temperature dependent [50].

Unfortunately, none of the above signals could be seen in our spectra (Figure 3). Water was a weakly coordinated ligand in the Au(III) complexes, and the ease of its substitution [49] may explain the impossibility of distinguishing it from bulk water. Another interpretation problem arises from the overlapping of H_3_, H_5_ and H_6_ [51].

However, the spectra of the reaction mixture (Figure 3b,d) and the control spectra of pure cyclodextrin (Figure 3a,c) are obviously different. In order to facilitate interpretation, we have compared our spectra to those of 26 structures of oxidized α-cyclodextrin, such as primary hydroxyls oxidized to aldehydes and to carboxyls, and secondary hydroxyls oxidized to diketones, calculated by means of an online predictor [52]. The spectra of two of them were additionally calculated using ChemDraw, which is part of ChemOffice Suite 2018 v18.0.0.231. However, none of the calculated spectra looked similar to Figure 3b or Figure 3d. The experimentally observed singlet of an aldehyde group of oxo- or dioxo-α-cyclodextrin at 9.7 ppm or a multiplet of aldehyde groups of randomly oxidized α-cyclodextrin at 9.85–9.51 ppm [53] were also absent from our spectra. Therefore, the complexation of α-cyclodextrin with gold ions is not accompanied by its oxidation, and the difference in spectra is caused by the complex formation itself, as shown by the stearic acid inclusion in γ-cyclodextrin that caused the de-shielding of all its protons [51]. The signal of the H_3_ proton of α-cyclodextrin also shifted after the encapsulation of a terbium complex with an aromatic ligand. Its coordinated water did not appear in the NMR spectrum in D_2_O [54]. After the encapsulation of gold nanoclusters, the same H_3_ proton became even more shielded than H_5_ and H_6_ [43]. This effect could be observed in our system as well (Figure 3d). Numerous changes in the chemical shifts of cyclodextrin protons along with the change in the H_5_ proton peak from a doublet to a singlet was ascribed to an inclusion-complex formation between the β-cyclodextrin-ionic liquid and 2,4-dichlorophenol [55]. The encapsulation of vanillin caused the shielding of the H_3_ and H_5_ protons of β-cyclodextrin [56]. After the synthesis of gold nanoparticles using β-cyclodextrin, its peaks also slightly shifted downfield [13]. However, another possible cause of this spectrum alteration, viz. the presence of paramagnetic [57] Au(II) intermediate, also cannot be ruled out.

The peaks near 5.5 ppm present in the control spectrum in DMSO (Figure 3c) but absent from all other spectra were assigned to hydroxyl groups [58]. More precisely, the downfield and upfield peaks in DMSO correspond to the hydroxyl groups of carbon atoms 2 and 3, respectively, while the triplet at 4.5 ppm correspond to the primary hydroxyl group [55].

The water peak that may represent both bulk and coordinated water shows more pronounced coupling with the protons adjacent to hydroxyls in ROESY (Appendix A) and NOESY spectra. This may lead to the conclusion that the gold ions are coordinated with cyclodextrin outside its cavity, as shown in unhydrolyzed [AuCl_4_]^−^ in KAuCl_4_ [45,59] and later in NaAuCl_4_ that forms a complex having another structure [45]. As in the above examples, the gold ions in our case were coordinated by hydrogen bonds because the degree of cyclodextrin ionization at pH below 12 was negligible [36]. However, our gold ions were different. At pH 6.16, the predominant anion was [AuCl(OH)_3_]^−^, although [AuCl_2_(OH)_2_]^−^ was present as well [30]. It should be noted that they actually exist as a cation [AuCl(OH)(H_2_O)_2_]^+^ and neutral species [AuCl_2_(OH)(H_2_O)], respectively [60].

Oxygen-17 has a chemical shift range of about 2000 ppm [42]. Therefore, we tried ^17^O NMR in the hope that it would allow us to resolve our complex water peak.

There is only a peak at 0.6703 ppm in D_2_O, and it corresponds to water [61]. The fast exchange of water bound to maltodextrin with bulk water was reported [61], making them to appear as one singlet. The peak shifted downfield compared to its position in the control spectrum at 0.27 ppm because of the presence of NaCl and residual HCl in the reaction mixture. Both of these chemicals shift water resonance downfield [62,63].

The spectrum in DMSO strongly resembles that reported for 10% water in DMSO [64], thus allowing easy peak assignment. The positions of the peaks in our case (4.3848 ppm for D_2_O and 14.7230 ppm for DMSO) were different from both the above report [64] and data for pure solvents (−3 ppm for D_2_O and 13 ppm for DMSO [65]) but this can be explained by the effects of hydrogen bonds. Hydrogen-bond formation causes de-shielding of the oxygen atom. Conversely, breaking hydrogen bonds leads to shielding [66]. Indeed, hydrogen-bond lifetimes between water and DMSO and even between water molecules in DMSO are 5 and 3 times more, respectively, than between water molecules in pure water. Water–DMSO bonds are also stronger than the bonds between only water molecules, although the average coordination number of molecules in a water–DMSO system is less than in pure water [67]. Therefore, the chemical shifts in DMSO–water mixtures should depend on water concentration, and in our experiment, it was much lower than previously analyzed, both by NMR [64] and theoretically [67]. In the control spectrum (without gold), only the DMSO singlet at 15.759 ppm could be seen. The absence of cyclodextrin signals can be explained by the low temperature of 22 °C because the resonances of glucose begin to disappear below 64 °C even in ^17^O-depleted water [68]. However, in our case, switching to elevated temperatures would strongly accelerate the redox reaction, thereby preventing such an experiment from being performed.

The XPS data (Figure 2) proved the presence of cyclodextrin on the particle surface by clear peaks of carbon and oxygen. Sodium (1.9%) was also found (Table 1), and was probably a counterion to the carboxyl groups of oxidized cyclodextrin. Chlorine (2.4%) was also found, which may have been from unreacted [AuCl_2_]^−^. Au4f_7/2_ and Au4f_5/2_ spin-orbit splitting caused appearance of two pairs of peaks in the Au4f core-level spectrum (Figure 2b). The positions of the primary pair correspond to elemental gold [69], and those of another pair at binding energies of 85.1 eV and 88.7 eV correspond to Au^1+^ [70]. The fitting resulted in 85.1% for Au(0) and 14.9% for Au^1+^ (Table 2). Some of the gold atoms were recognized as the gold-oxide-state of [AuCl_2_]^−^, while others may have been charged surface atoms neighboring the carboxyl groups of immobilized cyclodextrin [12].

The FTIR spectra of the synthesized gold nanoparticles and of β-cyclodextrin (Figure 4) coincide, thus providing direct proof of the successful immobilization of cyclodextrin onto their surface. The broad band of the hydroxyl groups reported at 3401 cm^−1^, C-H stretching reported at 2925 cm^−1^, H-OH bending reported at 1638 cm^−1^, glycosidic bridge antisymmetric stretching vibration reported at 1157 cm^−1^, coupled C-O stretching vibration at 1080 cm^−1^, coupled C-C stretching vibration at 1030 cm^−1^ [71], skeletal vibration involving α-1,4 linkage at about 945 cm^−1^ and ring ‘breathing’ vibration reported at 754 cm^−1^ [72] can be seen in both spectra. It should be noted that the peaks of the gold nanoparticles are shifted compared to those of β-cyclodextrin because of its interaction [71] with the particle surface.

The yield of gold nanoparticles calculated from absorbance spectra (Appendix A) was 6%. This is less than those reported for other systems (e.g., 75% for gold nanoparticles functionalized with a thiol containing a phosphate and a quaternary amino group [73]; 52.5% for those synthesized using *Curcuma mangga* extract [74]; 15% for those with RNA layer inside peptide-containing lipid envelope [75]). For the calculation of the latter yield [75], only the coating stage was considered, and the possible loss of gold during the synthesis itself was not taken into account. However, it should be underlined that the main goal of the present work was to ensure the complete absence of aggregates in order to provide a ready-to-use technique aimed at biomedical applications, but the final solutions usually contained aggregates. In order to have aggregate-free solutions, the samples should be withdrawn in the middle of the reaction, and this factor decreased the yield. Moreover, the above data are for centrifuged solutions. In order to prevent aggregation during centrifugation, mild conditions leading to the significant loss of nanoparticles were selected (Appendix A). Harsh centrifugation conditions would definitely increase the yield, but at the cost of partial aggregation. A third factor is the limitation of the yield-calculation technique itself. It assumes a monomodal and narrow size distribution, which is not always the case. Hence, the yield might be somewhat underestimated.

## 4. Conclusions

The formation of gold nanoparticles in the cyclodextrin–tetrachloroaurate system follows Finke–Watzky kinetics, although the real mechanism is more complex, and the resulting nanoparticles are mostly polycrystalline. The particle-size distribution is pH dependent. The preparations that were produced at optimal reaction conditions have high concentrations of nanoparticles and do not contain aggregates or other impurities. These optimal parameters are summarized in Appendix A. From these, 0.004 M cyclodextrin is preferable due to its better reproducibility. The synthesis can be carried out at room temperature without organic solvents in a solution containing only cyclodextrin, HAuCl_4_, and NaOH, thus satisfying green-chemistry criteria and meeting the demands of pharmaceutical applications.

## Figures and Tables

**Figure 1 nanomaterials-12-00583-f001:**
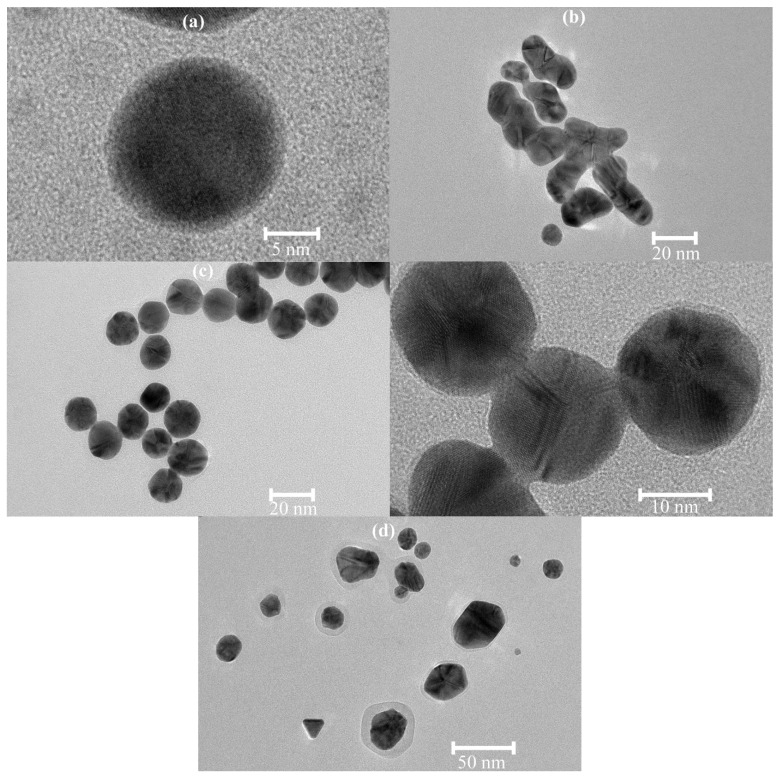
TEM images of gold nanoparticles synthesized at 0.1% HAuCl_4_ at room temperature: (**a**) at 0.002 M α-cyclodextrin, pH 11.5 in the presence of NaCl after 54 min; (**b**) at 0.004 M β-cyclodextrin, pH 11.5, after 44 min and (**c**) 2 h 17 min; (**d**) at 0.002 M α-cyclodextrin, pH 11, after 12 h 38 min and (**e**) 43 h 29 min; (**f**) at 0.004 M α-cyclodextrin, pH 11, after 18 min and (**g**) 6 h 38 min; (**h**) at 0.004 M β-cyclodextrin, pH 11, after 3 h 15 min; (**i**) at 0.002 M γ-cyclodextrin, pH 11, after 6 h 38 min and (**j**) 23 h 33 min.

**Figure 2 nanomaterials-12-00583-f002:**
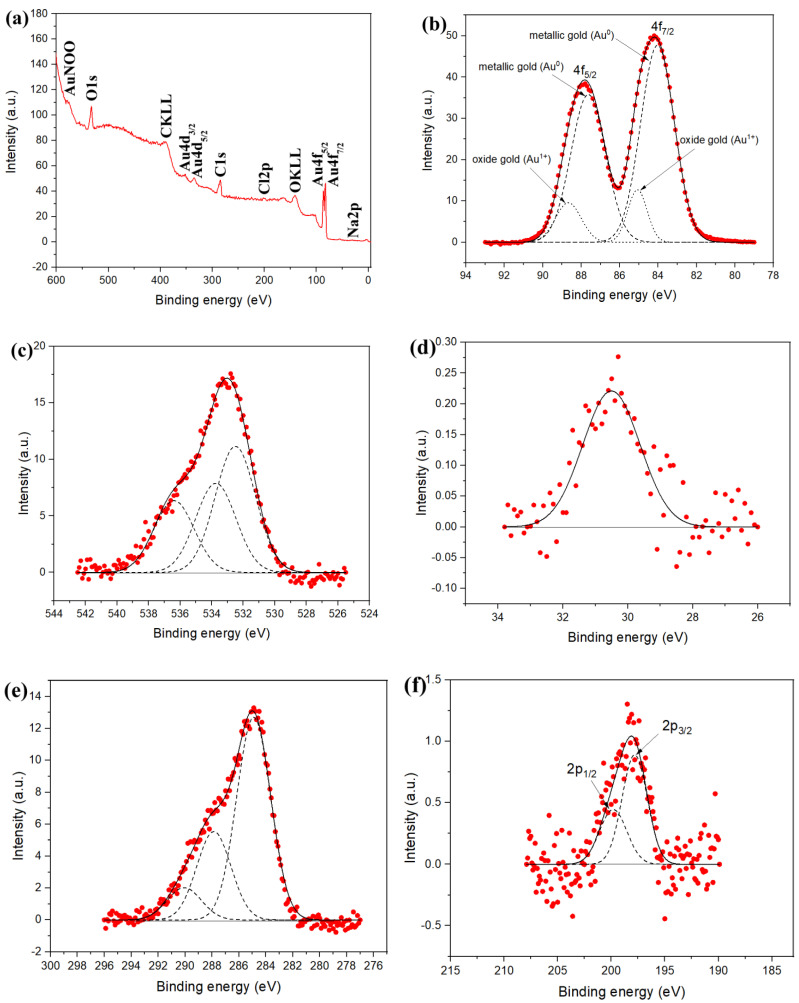
XPS spectra of gold nanoparticles: (**a**) an overview spectrum; (**b**) Au4f core-level spectrum; (**c**) O1s core-level spectrum; (**d**) Na2p core-level spectrum; (**e**) C1s core-level spectrum; (**f**) Cl2p core-level spectrum.

**Figure 3 nanomaterials-12-00583-f003:**
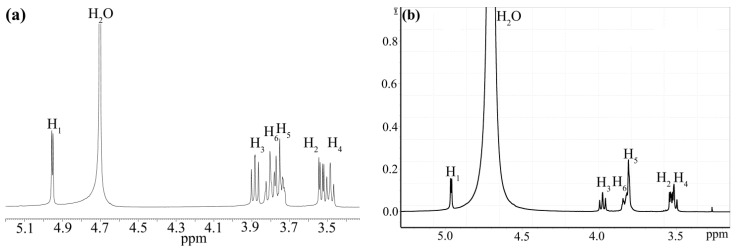
¹H NMR spectra of α-cyclodextrin alone (**a**,**c**) and in the presence of HAuCl_4_ at pH 6.16 (**b**,**d**) in D_2_O (**a**,**b**) or DMSO (**c**,**d**).

**Figure 4 nanomaterials-12-00583-f004:**
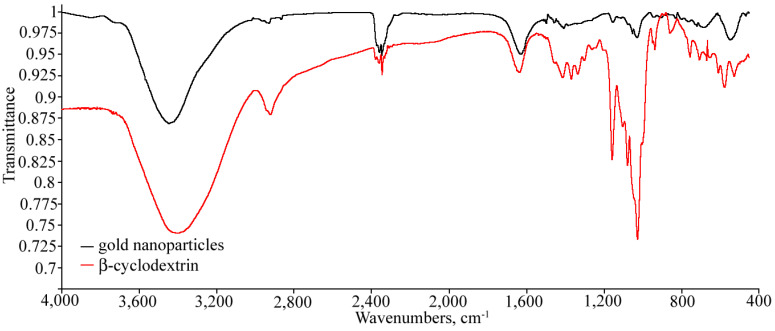
Normalized FTIR spectra of β-cyclodextrin Cavamax W7 and of gold nanoparticles with immobilized β-cyclodextrin synthesized under reflux conditions.

**Table 1 nanomaterials-12-00583-t001:** Elemental composition of gold nanoparticles according to XPS analysis.

Element Line	Atomic Percent
O1s	24.8
C1s	48.9
Au4f	22.0
Cl2p	2.4
Na2p	1.9

**Table 2 nanomaterials-12-00583-t002:** Fitting parameters for each component of the Au4f spectrum. Area is normalized by the atomic sensitivity factor.

	Au^0^		Au^1+^	
	7/2	5/2	7/2	5/2
Binding energy, eV	84.0	87.6	85.1	88.7
Full width at half maximum of the Gaussian peak	2.1	2.1	1.2	1.6
Area	1.25	1.22	0.19	0.24
Percentage	43	42.1	6.5	8.4

## Data Availability

Not applicable.

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
