# Peer review of "Formation of Aggregate-Free Gold Nanoparticles in the Cyclodextrin-Tetrachloroaurate System Follows Finke–Watzky Kinetics"

_nanomaterials, 2022, doi:10.3390/nano12040583_

Round 1

Reviewer 1 Report

Dear author,
my report file is attached. you can find my opinion about your paper.

Author Response

  1. The nanoparticles we have synthesized are intended for the use as drug delivery vehicles because they are cyclodextrin-capped. The goal of the present manuscript is to provide a ready-to-use synthesis technique and insight into the mechanism. Therefore, our manuscript falls into both "Synthesis" (green technology, functionalization, bioconjugation, and characterization) and "Medicine" (drug delivery) topics of this Special Issue. We hope now it will be obvious from the manuscript itself because of improved abstract, improved conclusions and three new sentences in the very end of the Introduction.
  2. Abstract has been partially rewritten, and absorbance and gold (I) content have been added.
  3. There are 4 references to Nanomaterials now. None of them has been added 'artificially': they were needed to improve the Results and Discussion section.

Reviewer 2 Report

The authors described the Formation of Aggregate-Free Gold Nanoparticles in the Cyclodextrin-Tetrachloroaurate System. Gold nanoparticles were synthesized with cyclodextrins at room temperature. The authors investigated the product by different analytical techniques and presented well. The obtained results are interesting for many readers. Before the manuscript will be accepted for publication the authors should arranged the tables because Table 3 has been add after the conclusion section, it should be corrected. 

Author Response

Table 3 has been moved to supplementary information and is now Table S1.

Reviewer 3 Report

I found the manuscript clearly written and easily understandable. It contains some new results which may be interesting to some Nanomaterials readers.  Therefore, I recommend publication of the manuscript after addressing the following comments:

  1. FT-IR analysis should be performed to detect the functional groups of gold nanoparticles.
  2. I would recommend comparing the yield of gold nanoparticles synthesis in this study with previously reported conventional methods.
  3. The aim and novelty of the research should be clearly stated at the end of the introduction section.
  4. There are many short subsections in the materials and methods part, which should be merged.
  5. The abstract needs improvement. It should be informative and completely self-explanatory.
  6. The conclusion should be improved. Table 3 can be provided as supporting information.
  7. SEM images will help to gain further information on the morphology of the synthesized nanoparticles.
  8. Following references should be cited to improve the quality of the paper:

Journal of hazardous materials 381 (2020): 120742. DOI: 10.1016/j.jhazmat.2019.120742

ACS Applied Materials & Interfaces, 13 (2021) 13072-13086. doi: 10.1021/acsami.0c21076

Separation and Purification Technology 261 (2021): 118274. DOI: 10.1016/j.seppur.2020.118274

Author Response

  1. The presence of functional groups of cyclodextrin onto the surface of the nanoparticles is now proven by FTIR spectrum.
  2. The yield of the reaction is compared with those from other studies published in Nanomaterials. The corresponding paragraph is in the very end of Results and Discussion.
  3. Three new sentences stating the aim and novelty of the present research were placed in the very end of the Introduction.
  4. Subsections 2.4, 2.7, 2.8 and 2.9 from the last version of the manuscript are now merged into subsection 2.6. (Now it contains FTIR technique as well).
  5. The abstract has been partially rewritten.
  6. Conclusions have been partially rewritten in order to become clearer.
  7. Supplementary information now contains SEM images (Figure S1).
  8. The references provided by the reviewer have been cited. The corresponding text is in the introduction. It has allowed to identify another possible field of the application of cyclodextrin-capped gold nanoparticles the present manuscript deals with: wastewater treatment.